# Sensitivity Analysis of Modelled Flood Inundation Extents over Hawkesbury–Nepean Catchment

S. L. Kesav Unnithan [1,2,3,4,*], Basudev Biswal [2], Wendy Sharples [4,*], Christoph Rüdiger [3,4], Katayoon Bahramian [4] and Jiawei Hou [4]

1   IITB-Monash Research Academy, Mumbai 400076, India
2   Department of Civil Engineering, IIT Bombay, Mumbai 400076, India
3   Department of Civil Engineering, Monash University, Clayton 3168, Australia
4   Bureau of Meteorology, Melbourne 3008, Australia
*   Correspondence: kesav.sreekuttanlakshmidevi@monash.edu (S.L.K.U.); wendy.sharples@bom.gov.au (W.S.)

**Abstract:** Rainfall runoff and topography are among the major factors controlling the accuracy of modelled riverine inundation extents. We have evaluated the sensitivity of both these variables on a novel 1-D conceptual flood inundation model employing Height Above Nearest Drainage (HAND) thresholds within sub-catchment units called Reach Contributing Area (RCA). We examined the March 2021 flood extent over the Hawkesbury–Nepean Valley (HNV) with 0.05' gridded runoff derived from the Australian Water Resources Assessment (AWRA) modelling framework. HAND thresholds were enforced within each RCA using rating curve relationships generated by a modelled river geometry dataset obtained from Jet Propulsion Laboratory (JPL) and by modelling Manning's roughness coefficient as a function of channel slope. We found that the step-like topographic nature of HNV significantly influences the back-water effect within the floodplain. At the same time, the improved accuracy of the GeoFabric Digital Elevation Model (DEM) outperforms SRTM DEM-derived flood output. The precision of HAND thresholds does not add significant value to the analysis. With enhanced access to river bathymetry and an ensemble point-based runoff modelling approach, we can generate an ensemble runoff-based probabilistic extent of inundation.

**Keywords:** flood inundation; HAND value-based threshold; Australian Water Resources Assessment; GeoFabric

## 1. Introduction

Hydrological models are distilled representations of real-world systems that estimate different components of the water cycle and water balance. They help decision-makers make informed choices for the planning and operation of water resources while considering the interconnections between physical, ecological, economic, and social components within a real-world system. Traditional hydrological models involve transforming rainfall observations into runoff/streamflow and Evapo-Transpiration (ET) estimates, upon considering catchment characteristics, including morphological features of river networks. The process usually involves complex calibration approaches of discharge and ET measurements for determining the weights of inherent catchment processes. Several types of hydrological models [1] varying in complexity exist for determining discharge; however, they cannot be used as-is in the case of flood management scenarios. This is because the discharge measurements need to be fed into hydrodynamic models, which then calculate the inundation height for every pixel/grid of interest within the catchment. The inundation units can be classified into a vector and raster-based units of inundation. The hydrodynamic models can also be categorised into 1D, 2D, and 3D flow models, with increasing computational costs in that order. Combining hydrological and hydrodynamic models, such as SWAT and LISFLOOD-FP [2], respectively, in operational situations due to their requirements for detailed information of the river system and resulting computational

costs is thus impractical for continuous modelling of flood inundation extents. Generally, design rainfall-runoff events are pre-processed, based on which the hydrodynamic models evaluate the inundation extents for designing and evaluating flood disaster recovery [3]. Such event-based or library lookup scenarios do not include the effects of antecedent climate and catchment characteristics or geomorphological processes, including erosion, which can accentuate the magnitude of design flood extents. Thus, this study focuses on simplifying hydrodynamic approaches without compromising flood extent accuracy. These approaches can complement continuously executing hydrological models without significantly increasing computational costs.

Further complexity in inundation modelling involves the data-scarce nature of river reach geometry. Currently, satellite missions can measure drainage width and length precisely. However, satellite signals cannot penetrate the water surface, and thus we cannot remotely measure the river depth and subsequent profile. Consequently, we depend on field surveys of river bathymetry to estimate the channel-carrying capacity. Several studies have used such river geometry data at point locations across the globe to model contiguous bankfull reach characteristics as a function of discharge [4–7]. We note that there is significant uncertainty in the modelled channel shape and the time-varying nature of channel morphology, which can result in divergent estimates of the bankfull carrying capacity of such rivers. Several studies [8–11] have analysed the impact of ensemble modelling approaches considering uncertainty in channel bathymetry, shape, and conveyance capacity for hydrodynamic systems implemented for specific sub-basins and catchments. For large-scale operational analysis of inundation extents, we need to deal with insufficient discharge measurements, unreliable drainage morphology, error-prone topographic datasets, uniform rainfall assumptions within catchment boundaries, and computational costs associated with the ensemble-based continuous execution of each of the processes dealing with the same.

Conventional methods to deal with large-scale inundation include generating static flood maps processed for designing flood rainfall events by running computationally complex hydrodynamic models in pre-emptive mode. However, running such models operationally across extensive spatial scales with near real-time data under the above constraints limits the analysis to adopting conceptual inundation modelling approaches. Height Above Nearest Drainage (HAND) is one such static topographic indicator of the inundation potential of a location obtained from the Digital Elevation Model (DEM) [12]. HAND is a hydrologically consistent indicator of local draining potential compared to other topographic indicators, including the Topographic Wetness Index (TWI) [13]. HAND value-based thresholding, where raster pixels within a catchment that have HAND values less than the observed gauge level are considered submerged, has been employed to map inundation across different spatial scales. One example is across the entire Amazon basin [12], subject to the quantity and quality of gauged data. Upon availability of river bathymetry and access to large computational resources, 1D, 2D, and 3D flow routing procedures, including Muskingham–Cunge [14], mizuRoute [15], CaMa-Flood [16], and even Regional Flood Frequency Approach (RFFA)-based methods [17] can be incorporated to estimate the flow rates needed for rating curve estimation of HAND thresholds.

Due to the unavailability of ground-surveyed river geometry data and the need for rapid flood mapping, we consider the Conceptual Flood Routing and HAND-Based Inundation Model (CFRHIM) (Unnithan et al., A novel conceptual flood inundation model for large scale data-scarce regions, submitted to Environmental Modelling and Software, 2022, henceforth referred to as submitted manuscript). CFRHIM has a modular approach that can be coupled with most hydrological models and executed for large time series. It overcomes error-prone topographic information from global access DEM and considers the modelled river bankfull width and depth estimate input. With the paucity of standardised river geometry datasets and the need to achieve improved computational costs across large spatial scales, we rely on the river bathymetry information from [18] to model the bankfull discharge estimation. CFRHIM models HAND-derived probability inundation extents for

each Reach Contributing Area (RCA) (Figure 1) as a function of synthetic rating curves that incorporate the uncertainty in modelled river geometry datasets.

To better understand the main sources of uncertainty in the CFRHIM modelling framework and in order to improve the overall accuracy of the CFRHIM, we evaluated the sensitivity of HAND-derived inundation extents to:

1.  The product quality of the gridded runoff datasets—AWRA V6 (baseline) and AWRA V7 (better quality),
2.  The quality of the Digital Elevation Model (DEM) datasets—Shuttle Radar Topography Mission (SRTM—not hydrologically conditioned) and Australian Hydrological Geospatial Fabric (GeoFabric—hydrologically conditioned), and
3.  Incorporation of hydrological model processes including:

a.  back-water effects,
b.  modelled Manning's roughness coefficient, and
c.  precision of HAND intervals derived from synthetic rating curves.

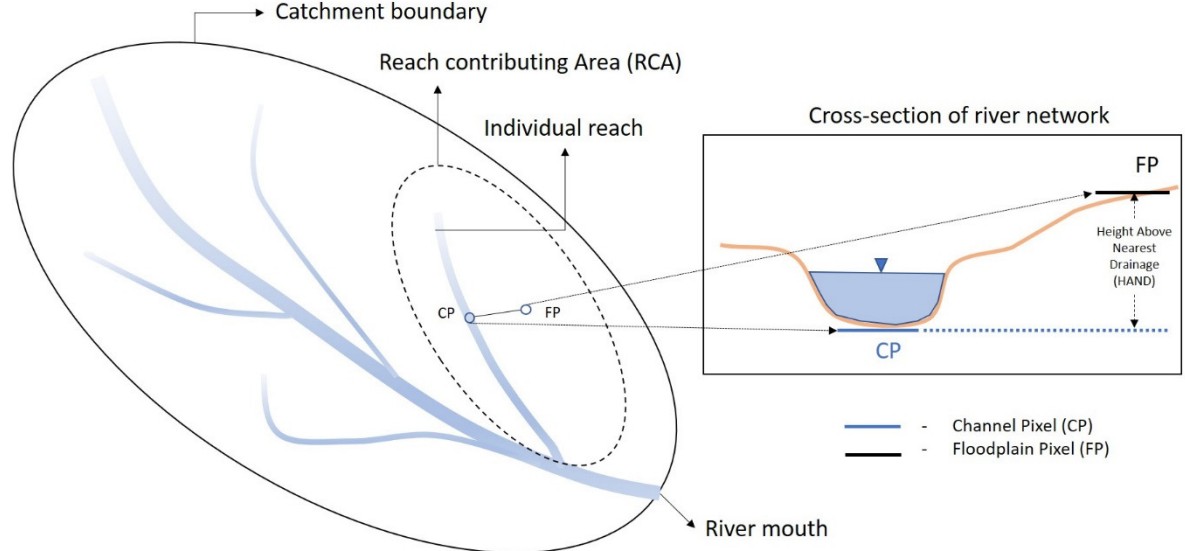

**Figure 1.** Conceptual representation of HAND index within the RCA.

A sensitivity study was performed for the Hawkesbury–Nepean Valley (HNV) catchment in New South Wales (NSW), Australia. Using this case study, we performed a sensitivity analysis to analyse the model uncertainties in mapping flood magnitudes for HNV. Specifically, we examined the nature of fluvial flooding patterns repeatedly observed in HNV and focused on identifying the critical datasets and model processes that control the accuracy of simple conceptual flood inundation extents. This study aimed to ascertain the best CFRHIM configuration, thereby evaluating the applicability of HAND-derived inundation extents to provide downstream applications of operational flood alerts for large-scale data-scarce regions.

## 2. Methodology

We used gridded runoff from AWRA-L (jointly developed by the Australian Bureau of Meteorology (BoM) and CSIRO) as an input forcing into CFRHIM. In addition, the GeoFabric DEM corrected from the NASA SRTM DEM at 30 m spatial resolution was used to model the HNV catchment topography [19]. The unique topographic environment of the HNV catchment presents the opportunity to examine the back-water effect (BE) of downstream inundation heights on upstream catchment RCA. The model structure of HAND value-based thresholding to derive inundation extents was subject to different precision intervals to understand the effect of synthetic discharge–flow height relationships

developed within each RCA (refer Figure 1). Furthermore, Manning's channel roughness coefficient, usually considered a constant value within a basin, was modelled as a function of the slope of the individual reaches [20]. The modelled roughness values were used to evaluate the probabilistic inundation extents for the 2021 HNV flood event.

### 2.1. Datasets Used for the March 2021 Flood over Hawkesbury–Nepean

The HNV catchment with an area of 22,000 sq km lies near Western Sydney in NSW in southeast Australia. The catchment is home to over 130,000 inhabitants [21] and the Wollondilly, Coxs, Grose, Colo, South Creek, and Macdonald tributaries, rich in riparian and mangrove vegetation. The Warragamba Dam is the major dam in the catchment, in addition to Wingecarribee, Avon, Cataract, Cordeaux, and Nepean dams, regulating flow for the water supply of Greater Sydney [22]. The region has an annual rainfall of 75 cm, with the January to March months over the last 100 years exhibiting high runoff and soil moisture retention capacity (Figure 2—right panel); however, it witnessed prolonged drought even after the Millennium Drought between 2010 and 2019 before it witnessed a 1 in 5-year flood event in 2020 [23]. The region again experienced a major 1 in 20-year magnitude flood event from 7–15 March 2021, causing 2 billion AUD in damages with one reported fatality [24]. We focused on the later flood event because we could readily compare model output with observed flood extents from Copernicus Emergency Management Services (CEMS). CEMS regularly publishes flood extent maps from available EU Sentinel satellite missions for different natural hazards, including flood events, volcanic activity, droughts, and forest fires. The 2021 HNV floods are catalogued as event EMSR504, with flood extent shapefiles publicly available [25].

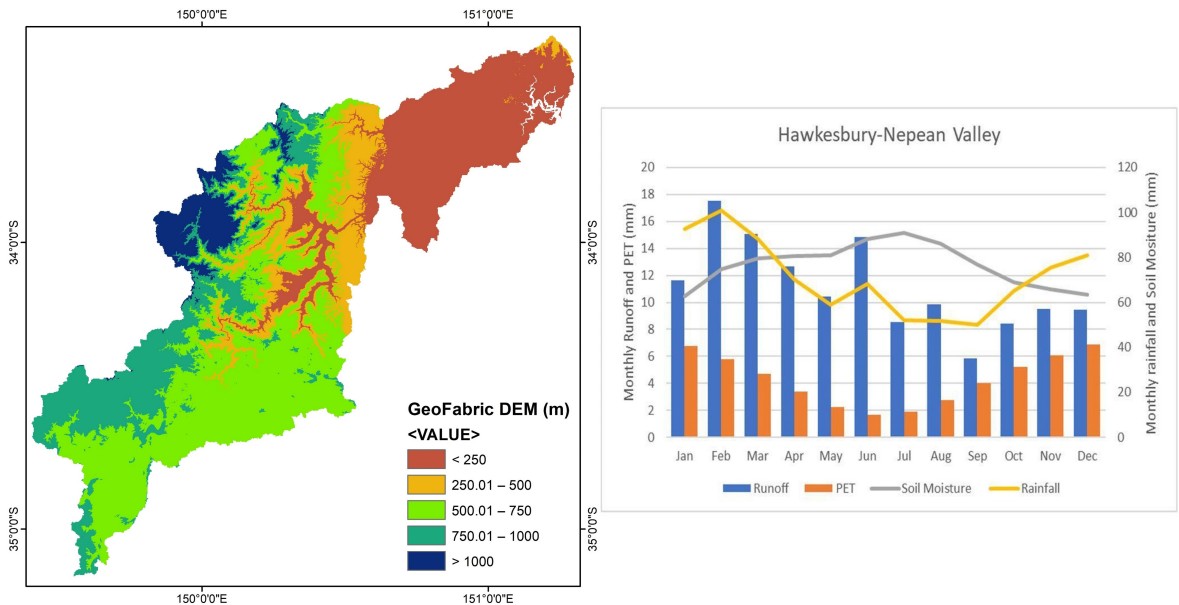

**Figure 2.** Hawkesbury–Nepean catchment 'near Sydney, New South Wales, Australia.

The Australian Landscape Water Balance Model, part of the BoM's operational Australian Water Resources Assessment Landscape (AWRA) modelling system's 0.05° × 0.05° gridded runoff product, Version 6 (AWRA-V6) and newer Version 7 [26,27], are used as input forcing for the CFRHIM inundation model. The inundation performance for model versions V6 and V7 was evaluated as part of the BoM's mandate for improved runoff estimates [26], with daily median Kling–Gupta Efficiency (KGE) and Nash–Scutcliffe Efficiency (NSE) upgraded to 0.48/0.50 in case of V7 from 0.43/0.49, respectively, for V6, analysed across 291 sites in Australia. Since flooding is a relatively short-duration phenomenon, improved modelled daily runoff estimates can significantly affect inundation dynamics. More information on version improvement is detailed in the methodology section. The

SRTM DEM [28] and the BoM's Australian Hydrological Geospatial Fabric (GeoFabric) DEM [19], both at 30 m spatial resolution, were used to derive the 2021 inundation extents. The GeoFabric DEM is derived from SRTM DEM with a stream network enforced at a scale of 1:25,000. The GeoFabric DEM was used to examine the performance of post-processing of native SRTM DEM to map flooding (Figure 2—left panel). Furthermore, the GeoFabric DEM is better resolved than the SRTM DEM, to the order of 0.001 m compared to the 1 m interval of SRTM. A step-based topography characterises the HNV catchment, as evidenced by GeoFabric DEM in Figure 2, with steep upstream channels followed by the floodplain in the Richmond–Windsor regions, with subsequent higher slopes further downstream. The catchment's stepped nature indicates the back-water effect's influence on inundation extents, which is explained in the following section.

### 2.2. Sensitivity Analysis of HAND-Based Inundation Extents

The schematic representing the process flow adopted in this study is shown below in Figure 3.

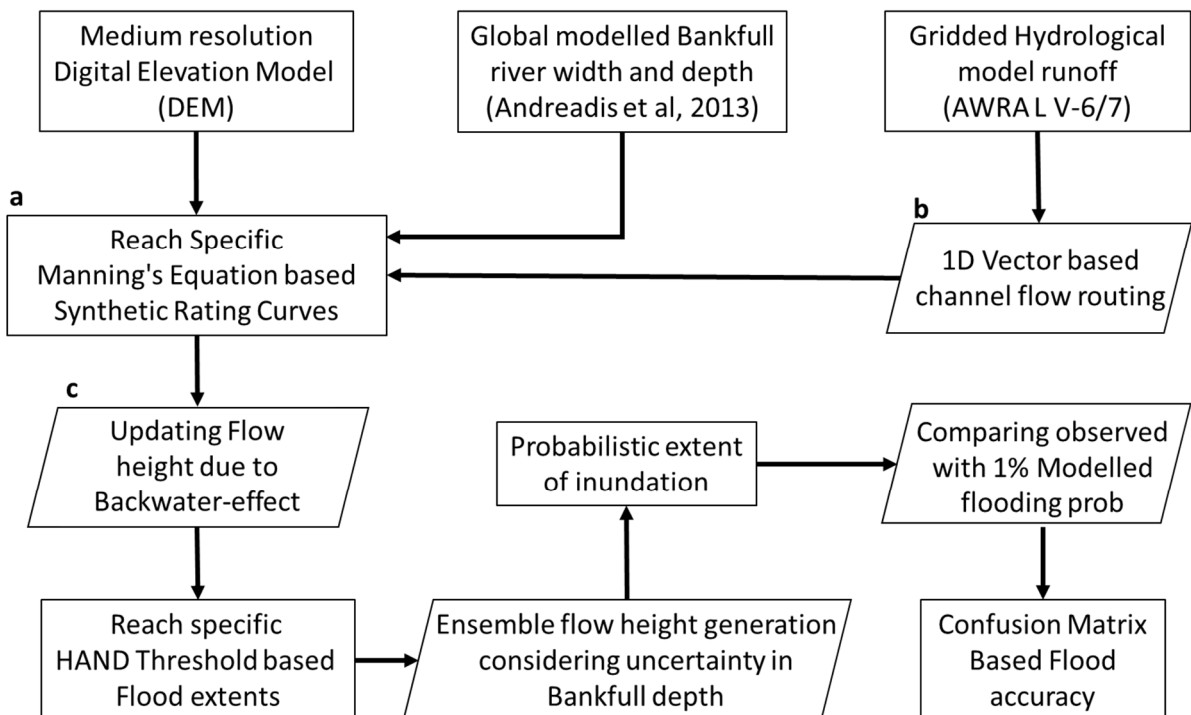

**Figure 3.** Flowchart depicting uncertainty in large-scale inundation analysis for data-scarce regions. (HAND: Height Above Nearest Drainage, AWRA L: Australian Water Resources Assessment—Landscape model Version 6/7) [18].

A modular approach towards mapping inundation was adopted to divide the region of interest into smaller hydrological units called RCAs. An RCA denotes the set of pixels that drain into a single reach of a channel network that does not branch. A HAND terrain map for each RCA is binarised using pre-defined flow-depth threshold values, and the corresponding extents are pre-processed and stored in a local lookup library. From Figure 3a, using the modelled river geometry datasets from [18]—NASA JPL, we generated the synthetic HAND-based rating curves using Manning's equation for each RCA in the channel network. The rating curves were generated such that overbank flow is considered within the channel by assuming a constant bankfull wetted perimeter for flow heights greater than bankfull depth, i.e., CFRHIM assumes the total flow occurs within the channel and that overbank flow only inundates laterally. Furthermore, Manning's roughness coefficient $n$ was modelled as a function of the slope and hydraulic radius of associated

reaches as given by [20]. The inundation extents were compared by keeping the roughness coefficient (*n*) constant at 0.03 (representative of a regular channel following the approach introduced by (Chow, 1959)) and by using the following equation,

$$n = 0.39 \times S^{0.38} \times r^{-0.16}$$ (1)

where *S* denotes the reach slope and *r* denotes the hydraulic radius. The above relationship was obtained by [20] for stable bed conditions in main natural channels with high slopes, even during large flow conditions. We evaluated the influence on channel flow upon assuming constant Manning's roughness co-efficient in data-scarce regions and by varying the same as a function of reach geometry in contrast to calibrating roughness as a function of observed inundated areas [11]. The area-weighted runoff for each RCA was then routed using the 1-dimensional flow routing procedure approach detailed in (Figure 3b—Unnithan et al., submitted manuscript). The approach considers the vectorised channel network scheme [15] used in delineating individual reaches and corresponding RCAs. The discharge is routed considering a spatially constant bankfull velocity to route runoff to the pour point (catchment outlet). Based on the rating curves derived for the river mouth reach, the constant velocity for the routed discharge is then assigned to all the reaches sharing the common pour point for subsequent discharge estimation. The ensemble of inundation depths is obtained from the synthetic rating curve for a given routed discharge. The rating curves consider uncertainty in modelled bankfull depth values, thereby generating an ensemble of flow heights and mapping the spatial extents and depths of inundation from the look-up library for each RCA. The flood extents derived within each RCA are then stitched across all the RCAs, representing the entire catchment area to obtain the comprehensive flood map.

The primary inputs to CFRHIM included the modelled river geometry datasets, the rainfall-runoff product, and topographic information. Model sensitivity to river geometry considers the uncertainty in river depth values affecting inundation extent to a greater degree than bankfull width and the shape of river reaches (Unnithan et al., submitted manuscript). In this study, we first examined the performance of HAND-derived inundation extents to different gridded runoff products by forcing AWRA V6 and AWRA V7 gridded daily runoff products for March 2021. The V7 runoff product involves improved static and dynamic inputs, including the height of the top of the vegetation canopy, spatial maps of tree basal area indicating maximum root water uptake instead of constant spatial value for the entire continent, and updated hydraulic conductivities [26,27]. Secondly, model sensitivity to topography was evaluated by comparing the highly resolved, hydrologically consistent, and ground-truthed GeoFabric DEM-derived flood extent to that derived from native SRTM DEM.

Thirdly, we examined the step-like topographic nature of HNV and conceptually represented the presence of the back-water effect (BE) (Figure 3c), i.e., the build-up of discharge in the downstream floodplain RCAs, which led to the stagnation of inundation in the upstream RCA. We initially estimated the ensemble of flow heights through the method described above. Consequently, the flow height at the terminal downstream reach outlet was propagated to the immediate upstream reaches by accounting for the difference in elevation of an upstream reach with the parent reach. This procedure was iteratively executed upon traversing upstream until the flow height of the parent reach was less than the elevation difference. Finally, due to the highly resolved nature of GeoFabric DEM, we also evaluated the precision of HAND thresholds in intervals of 0.25 m as compared to 1 m resolved SRTM DEM.

### 2.3. Validation

The modelled probability of inundation maps was compared with Sentinel-derived inundation maps published by CEMS in shapefile (shp) format. Since the comparison could be made only against available flooded/no flood (binary) observed datasets, the modelled inundation maps were reclassified into two categories by considering pixels

with a minimum 1% probability of inundation as being flooded. Here, the probability of inundation is the uncertainty in modelled overbank flood discharge-based extent. The resulting binary map was compared with the observed flood map using the confusion matrix (Table 1), considering the overlap of model flood/no flood (Mod Positive/Mod Negative) pixels with those of observed flood/no flood (Obs Positive/Obs Negative).

**Table 1.** Kappa confusion matrix used for comparison of modelled and observed flood pixels.

| Confusion Matrix | | Modelled Dataset | |
|---|---|---|---|
| | | Positive | Negative |
| Observed Dataset | Positive | True Positive (TP) | False Negative (FN) |
| | Negative | False Positive (FP) | True Negative (TN) |

We considered the following metrics based on the confusion matrix to evaluate the quality or success rate of the obtained inundation maps: Critical Success Index (CSI) [29], Youden's Index (YI) [30], and Cohen's Kappa (CK) (Cohen, 1960) [31], given by

$$CSI = \frac{TP}{TP + FN + FP} \tag{2}$$

$$YI = \frac{TP}{(TP + FN)} + \frac{TN}{(TN + FP)} - 1 \tag{3}$$

$$CK = \frac{2\,(TP \times TN\, -\, FN \times FP)}{(TP + FP)(FP + TN) + (TP + FN)(FN + TN)} \tag{4}$$

The CSI metric provides the relative accuracy of the model in capturing inundation patterns as a function of observed flooded and non-flooded pixels. A heavy penalty exists in the case of model over-estimation, wherein modelled extents exceed or do not precisely coincide with observed inundation. Thus, we consider a more holistic indicator in YI, which yields the optimum tradeoff between over- and under-estimation of flooding patterns. Cohen's Kappa is a similar spatial indicator which gives the overall agreement between two different sources of data sets. We report CSI, YI, and CK values for different model result comparisons to infer meaningful patterns captured by the model.

## 3. Results

We examine the effect of routing within the CFRHIM framework of HAND-based inundation techniques compared to directly mapping HAND extents from the gauged maximum flood discharge level measured at Windsor station in the Hawkesbury–Nepean floodplain. Significant over-estimation exists in the floodplain region in the case of directly mapping HAND-based inundation extent derived from a maximum flood level of 13.43 m across the entire catchment, gauged on 10 March 2021, as compared to the observed inundation for the same day provided by CEMS (Figure 4). We evaluate the flood maps in terms of the Critical Success Index (CSI), Youden's Index (YI), and Cohen's Kappa (CK). With reference to the bar plots in Figure 4, the CSI indicates high model bias showing over-estimation. For mapping extents directly based on HAND thresholds, we report a CSI of 0.25 for the HNV 2021 flood event. The YI is significantly higher at 0.58 since most of the flooded pixels are captured by the maximum level-HAND extents. The CK denotes the overall agreement between the extents, considering both flooded and non-flooded pixels; hence, we report a significantly low CK of 0.25.

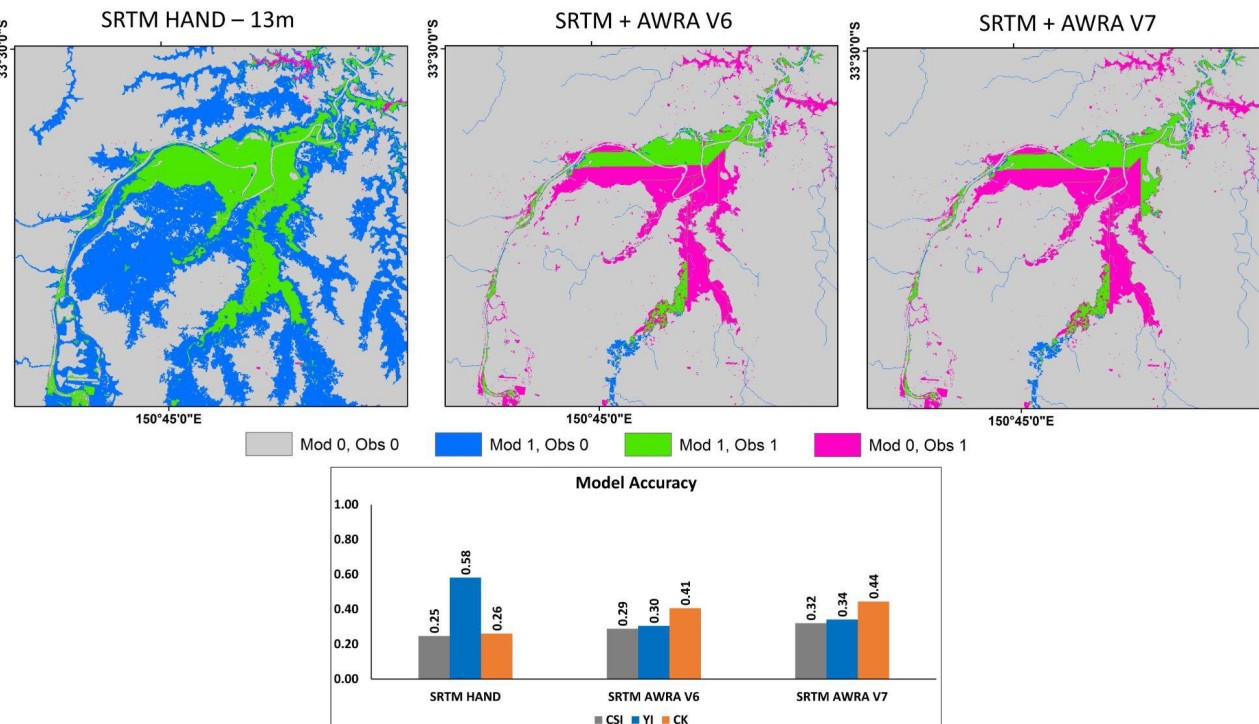

**Figure 4.** HAND-derived extents compared to CFRHIM-derived extents using SRTM 30 m DEM with AWRA V6 and V7 0.05° gridded runoff products.

In our first testing case, we investigate the sensitivity of SRTM DEM-based CFRHIM-derived inundation extents to different runoff products, including AWRA V6 and AWRA V7 (Figure 4). As demonstrated by a higher CK of 0.44 for V7 as opposed to 0.41 for V6, the V7 product with improvements in AWRA model parameterisation related to baseflow and reduced bias upon assigning 50% weightage to terrestrial water storage generates more runoff for the flood event in March 2021 than the V6 product. The inundation extents follow a similar pattern to the input runoff grids. Significant model underestimation exists because of erroneous SRTM DEM, which cannot distinctly delineate the floodplain from the channel pixels, resulting in a skewed channel network and, ultimately, poor modelled inundation accuracy.

Subsequently, in the second case, we analyse the sensitivity of CFRHIM extents to different DEM products; namely, SRTM and GeoFabric DEM at 30 m resolution with daily gridded AWRA V7 runoff input. The channel network delineated from both DEMs, as shown in Figure 5, indicates the improved reciprocation of channel morphology from GeoFabric DEM as compared to that derived from SRTM DEM. The ensuing CFRHIM-derived inundation maps depict the flood patterns following the GeoFabric-derived channel network more accurately than SRTM-based channel network. However, the performance evaluated by CK for CFRHIM with GeoFabric DEM and AWRA V7 runoff is significantly lower at 0.32 than that compared to SRTM (0.44). The poorer results can be attributed to the inability of synthetic rating curves to model flooding patterns for individual RCAs within the floodplain. We recognise the step nature of the catchment topography and hence focus on analysing the role of BE affecting inundation extents.

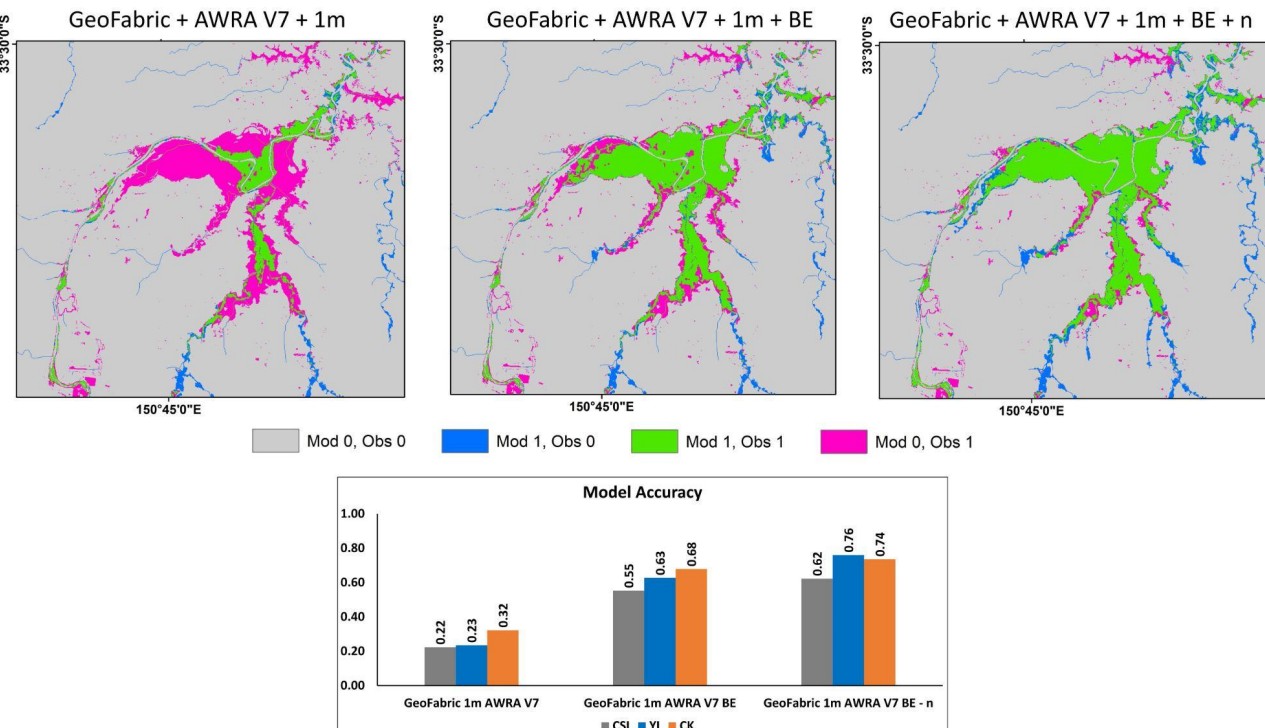

**Figure 5.** CFRHIM-derived extents using GeoFabric 30 m DEM with AWRA V7 0.05° gridded runoff for different model configurations, including precision of HAND threshold at 1 m intervals, back-water effect (BE), and modelled Manning's roughness (*n*).

In the third case, as explained earlier, we examine CFRHIM using GeoFabric DEM and AWRAV7 runoff input upon including model processes—BE (back-water effect) and modelled *n* (Manning's roughness). The modelled extents including BE significantly capture observed flood extents, thereby indicating the dominant effect of BE in flat floodplain regions that significantly affect inundation accuracy. The CK after including BE within CFRHIM improved to 0.68, a remarkable shift from when not including BE. By relying on more accurate DEM from the GeoFabric framework, we can reliably capture the BE on inundation extents. As explained in the methodology section, we also incorporated the modelled Manning's roughness coefficient *n* as a function of *S* and *r*. In this case, we report a higher CK of 0.74, thus denoting that flooding patterns are again crucially dependent upon the channel bed roughness in floodplain regions.

In the last case, we evaluate the sensitivity of the CFRHIM framework using the highly resolved GeoFabric DEM by considering the threshold of finer resolution in HAND values. The inundation results presented above consider a 1 m HAND interval between modelled extents derived from the synthetic rating curves described in the methodology section. The CFRHIM-derived extents with and without including BE and modelled *n* are as shown in Figure 6. The model results evaluated by CK show similar values and trends, with slightly better results in the case of 1 m HAND value thresholds. The CFRHIM model approximations in synthetic rating curve generation indicate better accuracy at coarser threshold resolutions, thus achieving accuracy without compromising computational costs/memory costs.

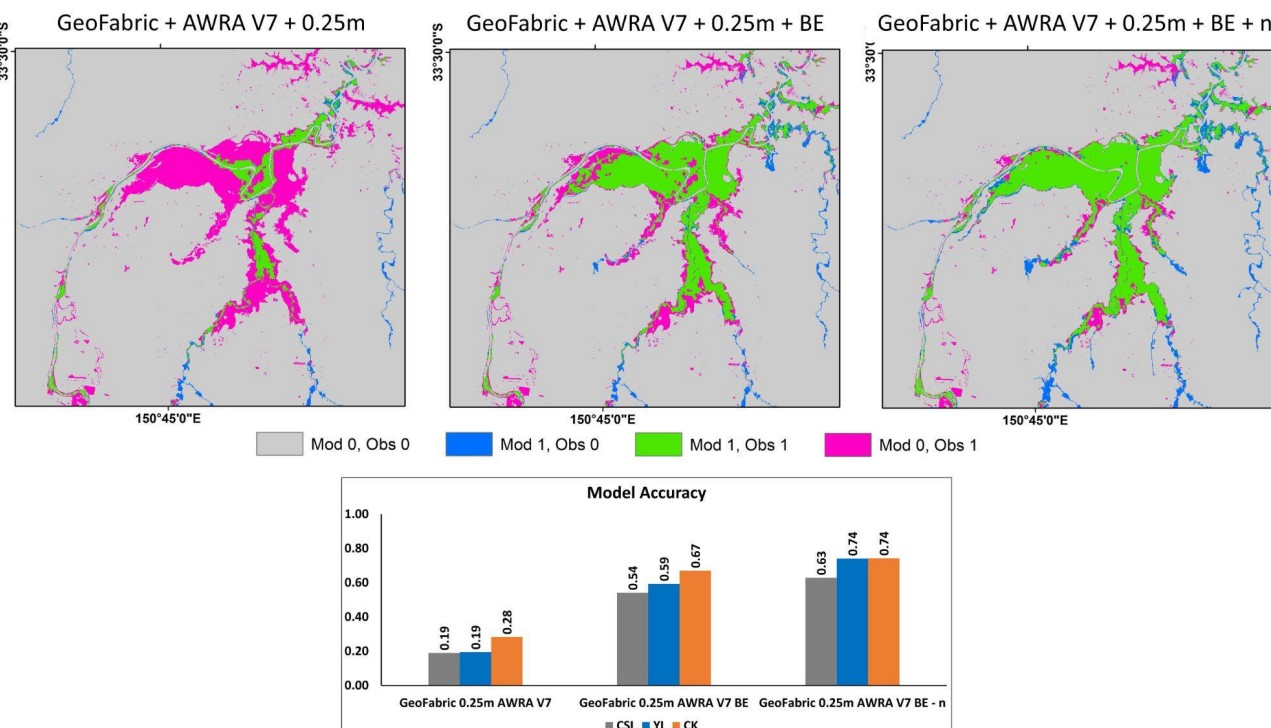

**Figure 6.** CFRHIM-derived extents using GeoFabric 30 m DEM with AWRA V7 0.05° gridded runoff for different model configurations, including precision of HAND threshold at 0.25 m intervals, back-water effect (BE), and modelled Manning's roughness (*n*).

## 4. Discussion

### 4.1. Effect of Runoff Estimations on Inundation Extents

Riverine floods are generally influenced by inflow volume to the channel network, characterised in modelling systems by routing available runoff into channel discharge. Hydrological models that capture gridded runoff accurately, as in the case of AWRA V7 compared to AWRA V6 [27], can help map flooding extents for large-scale data-scarce regions. The modelled extents can be improved with the availability of point-based discharge estimates, either gauged or modelled, which can be used in the calibration of routing schemas. Although uncertainty in routed discharge is investigated extensively for several short reaches and small-scale basins with fine-resolution river bathymetry, extensive discharge monitoring across large spatial scales during flood conditions might not be possible. Upon capturing associated inundation for a given flood discharge, we can more reliably compare with flooding patterns using remotely sensed imagery. Several conceptual flood mapping studies [32,33] have thus used HAND thresholds with stage–discharge relationships to map inundation but were severely hampered by poor peak flow estimates. A practical method for improving HAND-derived extents can be carried out by incorporating distributed ensemble discharge estimates [34] across several reaches within the basin of interest.

### 4.2. Topographic Conditioning of Flood Maps

In this work, we have greatly reduced the HAND-based overestimation of submerged floodplains by considering inundation within each RCA. This study provides a modular treatment of catchment areas, especially helpful in delta regions where even a minor error in flow height analysis can result in large floodplains erroneously mapped as inundated. The channel network delineation and ensuing RCAs are thus highly dependent upon the accuracy of input DEMs, the effect of which can be seen in the modelled inundated extents derived from SRTM and GeoFabric DEM. Channel and catchment topography can also affect flow within the catchment, evident in HNV with the back-water effect. In addition,

the traditional D8 channel delineation [35] adopted in this study omits the effect of deltaic regions on branching at the river mouth area. Additional topographic features, including hillslopes and aspect, must be considered to incorporate channel branching at river pour points. Furthermore, flood discharge can bring considerable sediment concentrations, altering the channel geometry. The availability of ground-surveyed reach geometry, even at specific gauge stations, can significantly improve model efficiency. Finally, evaluating spatial inundation extents for large-scale flood scenarios provides a more intuitive understanding of inherent model processes. Relying on sparsely observed discharge time series for evaluation is unreliable, especially under flooding conditions.

*4.3. Model Comparison with Observed Flood Maps*

Inundation extent comparisons with flood maps obtained from CEMS do contain inherent uncertainties in that the data fused from source satellite missions suffer from the presence of cloud cover in case of optical imagery, and radar back-scatter geometric and radiometric characteristics affecting inundated surface water retrievals, in case of SAR imagery. For the evaluation of CFRHIM-derived maps, based on prior studies, we convert the probabilistic maps into binary flood/no flood maps by considering pixels having more than a 1% probability of inundation as flooded. Nevertheless, the accuracy metrics reported follow the best evaluation schema currently in place for monitoring large-scale inundation extents [36]. The observed vector flood maps were resolved and compared at the same spatial resolution as the resultant modelled flood maps—at 30 m. Due to the lack of observed flood depth information, the modelled inundation depths could not be validated. The inundation model assumptions and approximations result in no improvement upon resolving HAND thresholds to higher precision. However, varying $n$ with $S$ and $r$ resulted in an improved CSI, YI, and CK for the 2021 flood event, which needs further investigation into other flood events over HNV and other flat catchment areas.

Based on the above inferences, we identify for HNV that AWRA V7 gridded runoff product resulted in more accurate inundation patterns being captured than upon using AWRA V6 and that the hydrologically conditioned GeoFabric DEM is best suited for capturing inundation dynamics in comparison to SRTM DEM. Overall, the sensitivity analysis resulted in the highest CSI, YI, and CK in the case of GeoFabric DEM-based CFRHIM-derived modelled inundation extents incorporating the BE and varying $n$ as a function of $S$ and $r$ at a coarser 1 m HAND threshold. The CFRHIM-derived extents significantly limit HAND-based over-estimation of floodplain inundation by considering individual RCA-based inundation depths. Relying only on HAND-derived extents that consider sparse water level thresholds (in this case, one gauge station) for the entire catchment during the flooding period results in significant overestimation (CK = 0.26). The GeoFabric DEM-based input results in a more realistic representation of floodplain inundation patterns than using SRTM DEM. This study focused on improvement of CFRHIM framework incorporating HAND thresholds for defining inundation extents and testing against observations. CFRHIM performance was not evaluated against the existing suite of data-intensive model approaches, which, although useful for intercomparisons, would require an independent study with in-depth analysis. CFRHIM model assumptions of constant flood velocity result in no improvement in captured inundation upon resolving HAND thresholds to finer precision, implying further scope for improvement of flood process characterisation.

## 5. Summary

We have evaluated the sensitivity of a HAND-based conceptual inundation model, CFRHIM, to different input datasets, including gridded runoff products (AWRA V6 and AWRA V7), topography (SRTM and GeoFabric DEM), and different model processes (BE, $n$, and precision of HAND-based synthetic rating curves). We observe that AWRA V7 performs comparatively better than AWRA V6 in mapping inundated areas due to higher gridded runoff values from V7. The modelled floodplain area significantly follows the river network upon using GeoFabric DEM compared to native SRTM DEM, especially useful

in catchments dominated by flat floodplains. The step-like topographic nature of HNV contributes to inherent flood processes with the significant presence of the back-water effect. Variation of channel roughness coefficient as a function of channel slope and hydraulic radius resulted in improved modelled extents. No such improvement in model accuracy was noticed upon adopting finer precision of HAND thresholds. We report the best CK of 0.74 in the case of GeoFabric DEM and AWRA V7 gridded runoff input considering BE and modelled *n* at 1 m HAND threshold intervals. The model performance for the 2021 HNV flood event was satisfactory, especially considering that this study did not include any ground-observed river geometry and gauged discharge datasets. There is further scope for incorporating an ensemble of point-based or gridded runoff products into the CFRHIM modelling framework for specific catchment scale inundation analysis. We emphasise and reiterate the need of the hour for the generation of an ensemble of inundation maps for probability analysis considering uncertainty in runoff and flow depth that can identify priority-based disaster preparedness and management policies.

**Author Contributions:** Conceptualization, S.L.K.U., B.B. and W.S.; methodology, S.L.K.U., B.B., C.R. and W.S.; investigation, B.B.; validation, S.L.K.U., W.S. and J.H.; formal analysis, W.S. and K.B.; writing—original draft preparation, S.L.K.U. and K.B.; writing—review and editing, S.L.K.U., B.B., C.R., W.S. and J.H.; supervision, B.B., C.R. and W.S.; project administration, W.S. and K.B.; funding acquisition, W.S. All authors have read and agreed to the published version of the manuscript.

**Funding:** This research received no external funding.

**Data Availability Statement:** Datasets utilized in the study available upon request.

**Acknowledgments:** S.L.K.U. expresses sincere gratitude to members of Hydrological Applications Team, Science Innovation Group, Bureau of Meteorology for their competent support in carrying out the current work.

**Conflicts of Interest:** The authors declare no conflict of interest.

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
