# Peer review of "Sensitivity Analysis of Modelled Flood Inundation Extents over Hawkesbury–Nepean Catchment"

_geosciences, doi:10.3390/geosciences13030067_

Round 1

Reviewer 1 Report

The subject of the paper is of great interest, but the authors should consider reviewing the way of presenting the information. The material is not always clear for a reader.

The last paragraph of the introduction (lines 104 to 119) seemed to be an explanation of the methodology. Maybe it could be moving in the methodology section.

As stated, it is not clear what the objectives of the study, this could be clarified in the Introdcution section.

Équation (1), line 186, is lacking some explanation. We could benefit from adding details to the use of the equation.

On lines 232-233, the authors explain that they will use the Cohen Kappa matrix, but later in the text, the use of two other indexes are added. This should be detailed in the paper. It is not clear how those three indexes are used to validate the results.

The result and discussion sections should be revised to better integrate analysis of the results. It is not clear which methodology of modelled flood is best, and which one should be used in which circumstances.

The subject is of great interest, but the paper should be revised to add clarity.

Reviewer 2 Report

See my comments attached!

Round 2

Reviewer 2 Report

Good job!